# Degradation Characteristics of Carbon Tetrachloride by Granular Sponge Zero Valent Iron

**DOI:** 10.3390/ijerph182312578

**Published:** 2021-11-29

**Authors:** Xueqiang Zhu, Yuncong Li, Baoping Han, Qiyan Feng, Lai Zhou

**Affiliations:** 1School of Environmental Science and Spatial Informatics, China University of Mining and Technology, Xuzhou 221116, China; fqycumt@126.com (Q.F.); zhoulai99@cumt.edu.cn (L.Z.); 2Department of Soil and Water Sciences, Tropical Research and Education Center, University of Florida, Homestead, FL 33031, USA; yunli@ufl.edu; 3School of Geography & Geomatics and Urban-Rural Planning, Jiangsu Normal University, Xuzhou 221116, China; bphan@cumt.edu.cn

**Keywords:** granular sponge iron, carbon tetrachloride, reductive dechlorination, chloroform

## Abstract

Granular sponge zero valent iron (ZVI) was employed to degrade carbon tetrachloride (CCl_4_). The effects of acidic washing, initial solution pH, and ZVI dosage on CCl_4_ degradation were investigated. Results showed that CCl_4_ was effectively removed by ZVI and approximately 75% of CCl_4_ was transformed into chloroform through hydrogenolysis. The rate of chloroform transformation was slower compared to that of CCl_4_, resulting in chloroform accumulation. CCl_4_ degradation was a pseudo first-order process. The observed pseudo first-order reaction rate constant (*k_obs_*) for CCl_4_ and chloroform were 0.1139 and 0.0109 h^−1^, respectively, with a ZVI dosage of 20 g/L and an initial CCl_4_ concentration of 20 mg/L. Surface acidic washing had a negligible effect on CCl_4_ degradation with ZVI. The *k_obs_* for CCl_4_ degradation increased linearly with increasing ZVI dosage and the optimal dosage of ZVI was 20 g/L based on the surface area-normalized rate constants. The negative relationship between *k_obs_* and the solution pH indicated that the degradation of CCl_4_ by ZVI performed better under weakly acidic conditions.

## 1. Introduction

Carbon tetrachloride (CCl_4_) has been used for a variety of purposes, including as a dry-cleaning solvent, metal degreaser, pesticide, refrigerant, fire extinguisher, flame retardant, and intermediate for industrial products [1]. Massive use, improper disposal and emission of CCl_4_ have caused severe soil and groundwater pollution [2,3]. CCl_4_ is the most common soil–groundwater contaminant [4], which poses a serious threat to ecological and human health and is found in at least 430 of the 1662 NPL sites [5]. In the 200 West Area of the U.S. Department of Energy Hanford Site, as much as 580 m^3^ CCl_4_ may have been released into groundwater [6]. CCl_4_ is a potential human carcinogen and has been identified as a priority pollutant by the Ministry of Environmental Protection of China, U.S. Environment Protection Agency and the European Commission [7]. The presence of CCl_4_ in groundwater is a major concern due to its toxicity, environmental persistence and widespread presence [8]. Therefore, remediation of CCl_4_-contaminated groundwater is of great significance.

Zero valent iron (ZVI) is an excellent electron donor capable of reductive transformation of contaminants with *E_h_*^0^ greater than −0.44 V (Fe^0^→Fe^2+^ + 2e^−^, *E*^0^ = −0.44 V) [9]. ZVI has been widely applied to remediation of sites contaminated by chlorinated hydrocarbons, heavy metals, nitrate and polycyclic aromatic hydrocarbons [10]. Compared to microscale iron particles, nano zero valent iron (nZVI) has a higher reactivity towards a variety of contaminants. The increase in the reactivity of nZVI particles mainly results from the increased reactive surface areas [11]. However, the reactivity of ZVI highly depends on the amount of reactive surface area since the degradation of contaminants by ZVI is a surface area reaction; the specific surface area normalized reaction rate constants for chlorinated hydrocarbons degradation with microscale zero-valent iron (mZVI) particles are in the same order of magnitude as obtained with nZVI particles, while mZVI corrosion rate is much slower [12]. In addition, mZVIs removed chlorinated aliphatic hydrocarbons more effectively than nZVIs when the ZVI material was used singly for field applications according to 112 field case studies provided in the literatures [13]. Recently, less expensive mZVI particles have been developed as a substitute for nZVIs to effectively degrade chlorinated hydrocarbons because of the high cost of nZVI and shortened lifetime due to high surface reactivity and fast corrosion rate with water [13,14,15]. The objective of this work was to evaluate the effectiveness of the highly porous sponge ZVI micro-particles in reducing CCl_4_ under laboratory conditions. The impacts of various parameters, including acid washing, ZVI dosage, and initial solution pH on CCl_4_ reduction were evaluated. Changes in concentrations of CCl_4_ and the intermediate products were monitored to identify the reaction mechanism.

## 2. Materials and Methods

### 2.1. Experiment Procedure

Irregular-shaped sponge iron particles used in this work were purchased from Tianjin Zhongcheng Iron Powder Factory (Tianjin, China). The material had a mean grain size less than 150 µm and a specific surface area of 0.078 m^2^/g [16]. The composition of the ZVI including Fe (96.52%), O (2.4%), Si (0.5%), Mn (0.29%), Ca (0.13%), Cr (0.048%), Mg (0.043%), P (0.038%), and S (0.031%) was analyzed using S8-TIGER X-ray Fluorescence (Bruker Corporation, Germany). The sponge iron particles (0.5 kg) were washed with 1.5 L of 0.1 mol/L H_2_SO_4_ in a 2 L beaker by mixing and stirring at 300 r/min for 0.5 h, and then rinsed with 1.5 L of ultrapure water for 0.5 h for five times. CCl_4_ was analytical reagent grade and purchased from the Sinopharm Chemical Reagent Company (Shanghai, China).

Batch experiments of CCl_4_ degradation by ZVI were conducted on a rotary shaker of 200 (±5) r/min at 25 (±0.2) °C. Acid-washed and non-acid-washed ZVIs (2 g) were placed in 100 mL serum vials with 100 mL deionized water, and 100 μL of 2 × 10^4^ mg/L CCl_4_ was added to form the test solution with the initial CCl_4_ concentration of 20 mg/L. The 2 × 10^4^ mg/L CCl_4_ solution was prepared by dissolving 12.5 μL CCl_4_ in 1 mL methanol and diluting to 10 mL with deionized water. Samples were taken at reaction times of 1, 4, 8, 16, 24, 48, 72 and 96 h for the analysis of CCl_4_ and its degradation products concentrations.

### 2.2. Analysis and Data Processing

The mass concentrations of CCl_4_, CHCl_3_, and CH_2_Cl_2_ were analyzed using an Agilent 6890N Gas Chromatograph (Agilent Technologies Inc., Santa Clara, CA, USA) equipped with a micro-cell electron capture detector and a G1888 Headspace Sampler and a 30 m long HP-5 capillary column. The temperature conditions were: oven temperature of 30 °C, injection port temperature of 150 °C, detector temperature of 250 °C. Separation was conducted with a temperature program: initial oven temperature 30 °C hold for 1 min, then ramped at 1 °C/min to 80 °C and hold for 1 min. Ultrapure nitrogen was used as a carrier gas for the GC with a flow rate of 2 mL/min (split ratio 10:1).

The surface morphology of ZVI was analyzed by a Quanta 250 environmental scanning electron microscope (FEI, Hillsboro, OR, USA) equipped with a QUANTAX 400-10 electric refrigeration spectrometer (Bruker, Germany).

The kinetics of the reductive dechlorination of CCl_4_ by ZVI can be described by a pseudo first-order reaction kinetic model [17].
(1)$$\frac{{dC}_{CT}}{dt}{=-k}_{obs}C_{CT}$$
where *C*_CT_ is the concentration of CCl_4_ at a certain time and *k_obs_* is the observed pseudo first-order reaction rate constant.

Studies showed that the reductive dechlorination rate constant is usually insensitive to changes in the initial concentrations of chlorinated hydrocarbons (e.g., trichloroethene concentrations of 1.21–61 mg/L) [18]. Therefore, the pseudo first-order reaction kinetic model is applicable to the reductive dechlorination of chlorinated hydrocarbons with concentrations within a typical concentration range.

The reductive dechlorination of CCl_4_ by ZVI is a surface reaction, and the dechlorination rate of CCl_4_ depends on the effective surface area of ZVI. Therefore, the reductive dechlorination kinetics of CCl_4_ can also be expressed by the surface-area-normalized rate constant [19].
(2)$$\frac{{dC}_{CT}}{dt}{=-k}_{SA}a_{s}\rho_{m}C_{CT}{=-k}_{SA}\rho_{a}C_{CT}$$
where *k_SA_* is the surface-area-normalized rate constant (L/(h·m^2^)), *a_s_* is the specific surface area (m^2^/g) of ZVI, *ρ_m_* is the mass concentration of ZVI (g/L), and *ρ_a_* is the surface area concentration of ZVI (m^2^/L). The *k_SA_*, calculated using the equation *k_SA_ = k_obs_*/*ρ_a_*, is independent of ZVI, the mass, specific surface area, and volume of the reaction solution. Due to its small range of variation, *k_SA_* is more suitable for the characterization of ZVI reactivity and the comparison of reactivity between different materials, compared with *k_obs_*. Equation (2) shows that the pseudo first-order reaction rate constant increases with the specific surface area of the material, which also means that the performance of the material can be improved by reducing the size of the material at the limited reaction range. However, as fine particulate materials typically possess low permeability, a balance has to be struck between reactivity and permeability in filed applications.

## 3. Results and Discussion

### 3.1. Effect of Acid-Washing Pretreatment on CCl_4_ Removal

Theoretically, oxides can be removed from the surface of ZVI particles through acid-washing, which increases the surface area of ZVI particles and facilitates the reductive dechlorination of chlorinated hydrocarbons by ZVI particles [20]. However, experimental studies show that pretreated ZVI is less effective in Cr(VI) removal as compared to untreated ZVI. This may be due to the fact that acid-washing leads to more severe mineral precipitation on the ZVI surface, thereby reducing the reactivity and long-term effectiveness of ZVI [21]. In our experiment, the reductive dechlorination of CCl_4_ by acid-washed and non-acid-washed ZVI was compared (Figure 1). Results showed that the relative concentration of CCl_4_ in the blank control group was 93.7% after 48 h and 90.1% after 72 h. The small reduction may be caused by volatilization. In addition, ZVI has low adsorption capacity and selectivity for targeted contaminants, especially for organic contaminants [22]. Xin et al. found that the amount of TCE removed by adsorption of XG-mZVI (the specific surface area is 0.136 ± 0.024 m^2^/g) was 1.80~2.08 µg and the amount of TCE removed by chemical reduction was 94.55~127.50 µg after 480 h reaction under different conditions [23]. The adsorption capacity of ZVI increases with the decrease in the particle size and the specific surface area of mZVI used in our study was only 0.078 m^2^/g. Therefore, it is believed that the adsorption of CCl_4_ on mZVI in our study can be ignored. In subsequent experiments, the loss of CCl_4_ caused by volatilization and adsorption on ZVI particles was ignored. The efficiency of CCl_4_ degradation by acid-washed ZVI was similar to that of non-acid-washed ZVI, with the respective rates of CCl_4_ removal being 87.4% and 82.7% after 16 h, and 96.6% and 95.9% after 24 h. These results indicated that acid-washing pretreatment did not significantly change the reductive dechlorination capability of ZVI on CCl_4_, and the respective observed pseudo first-order reaction rate constants under experimental conditions were 0.1167 h^−1^ and 0.1139 h^−1^.

The purpose of acid-washing was to remove the oxidation film on the ZVI particle surface. As shown in Figure 2, SEM results indicated that non-acid-washed ZVI particles had rough surfaces with a large number of pores, while acid-washed ZVI particles had smoother surfaces. The ZVI surface structure did not change significantly after acid-washing, and the EDS results indicated the absence of O (or the O content was below the detection limit) for both acid-washed and non-acid-washed ZVI. Therefore, it can be deduced that ZVI particles had a lower oxide coverage before and after acid-washing, which indicated that acid-washing pretreatment had no significant effect on the efficiency and rate of the reductive dechlorination of CCl_4_. Hence, in subsequent experiments, ZVI was unwashed.

During the degradation of CCl_4_, only chloroform (CHCl_3_, CF) was detected, while dichloromethane was not detected. As shown in Figure 1, the CHCl_3_ concentration increased gradually as CCl_4_ concentration decreased, with peak concentration occurring 24 h after the start of the reaction. When the degradation of CCl_4_ was complete, the CHCl_3_ concentration began to decrease.

### 3.2. Reaction Kinetics of Reductive Degradation of CCl_4_ by ZVI

Figure 3 displays that CCl_4_ degradation can be represented by a consecutive first-order reaction process. The concentrations of CCl_4_ and CHCl_3_ at any time can be expressed using Equations (3)–(7) [24,25]. (3)$${CCl}_{4}\;\text{degradation}:{{[CCl}_{4}]}_{t}{=[CCl}_{4}{]}_{0}e^{{-k}_{1}t}$$
(4)$${CHCl}_{3}\;\text{formation}:{CCl}_{4}\overset{k_{2}}{\to }{CHCl}_{3}{+Cl}^{-}$$
(5)$${{[CHCl}_{3}]}_{t}{=\alpha [CCl}_{4}{]}_{0}{\;(1-e}^{{-k}_{1}t});\;\alpha =\frac{k_{2}}{k_{1}}$$
(6)$${CHCl}_{3}\;\text{degradation}:\frac{{d[CHCl}_{3}{]}_{t}}{d_{t}}{=\alpha k}_{1}{{[CCl}_{4}]}_{0}{-k}_{3}{{[CHCl}_{3}]}_{t}$$
(7)$${CHCl}_{3}\text{concentration}:{{[CHCl}_{3}]}_{t}=\frac{k_{2}{{[CCl}_{4}]}_{0}}{k_{3}{-k}_{1}}\;\left(e^{{-k}_{1}t}{-e}^{{-k}_{3}t}\right)$$
where [*CCl*_4_]*_t_* and [*CHCl*_3_]*_t_* are the concentrations of CCl_4_ and CHCl_3_ at time t; [*CCl*_4_]_0_ is the initial concentration of CCl_4_; *k*_1_ is the overall pseudo first-order reaction constant for CCl_4_; *k*_2_ is the formation constant for CHCl_3_; *k*_3_ is the transformation constant for CHCl_3_; and *α* is the fraction of CCl_4_ transformed to CHCl_3_.

Table 1 and Figure 2 show the fitting results of the observed pseudo first-order reaction of CCl_4_ and CHCl_3_ degradation by ZVI. It was obvious that the CHCl_3_ removal capacity in the ZVI-H_2_O system was significantly lower than the CCl_4_ removal capacity. The redox potential of CHCl_3_/CCl_4_ (*E*^0^ = 0.673 V) is higher compared to the CH_2_Cl_2_/CHCl_3_ (*E*^0^ = 0.560 V) [26], which indicates that highly chlorinated aliphatic hydrocarbons are more easily reduced than species with a lower number of chlorine atoms [27]. Thus, the lower chlorinated hydrocarbons such as CHCl_3_ may accumulate in CCl_4_ plumes. The observed reaction rate constant for CHCl_3_ was approximately 10% as compared to CCl_4_, meaning that CHCl_3_ existed in the ZVI system for longer.

The ratio of CCl_4_ transformed to CHCl_3_ was less than 1.00 (0.7980 and 0.7569, respectively), indicating the presence of other competing reactions during the reductive dechlorination of CCl_4_, such as reductive hydrolysis that generates dichlorocarbene or chloroform free radicals. This is a characteristic of the CCl_4_ reductive dechlorination reaction [28].

### 3.3. Effect of ZVI Dosage on Reductive Dechlorination of CCl_4_

Figure 4 shows that CCl_4_ removal efficiency increases with the increase of ZVI dosage. After 16 h, the CCl_4_ removal rate in the system with a ZVI dosage of 5 g/L was only 42.2%, while the removal rate was 95.4% when the ZVI dosage was 40 g/L. After 48 h, CCl_4_ removal rates with ZVI dosages ranging from 5–40 g/L were 79.6%, 98.7%, 99.5%, 99.5%, and 99.5%, respectively, indicating that CCl_4_ could be completely removed when the initial concentration of ZVI exceeded 10 g/L. The reductive dechlorination of CCl_4_ by ZVI is a surface-mediated reaction; therefore, the dechlorination rate of CCl_4_ strongly depends on the effective surface areas. The number of effective active reaction sites in the ZVI-H_2_O system changes with ZVI dosage. A higher dosage of ZVI can provide more available reactive sites for CCl_4_ dechlorination and subsequently enhance the degradation of CCl_4_ [29].

Figure 5 demonstrates the reaction kinetics of the reductive dechlorination of CCl_4_ at different ZVI dosages (the relevant parameters are not listed). When the ZVI dosages were 5, 10, 20, 30, and 40 g/L, the observed pseudo first-order reaction rate constants (*k_obs_*) of CCl_4_ degradation were 0.0325, 0.0649, 0.1058, 0.1437, and 0.1788 h^−1^, respectively. Results indicate that *k_obs_* increased linearly with ZVI dosage, with the trend line fitted to Equation (8).
*k_obs_* = 0.0041*ZVI_dosage_* + 0.0194, *R*^2^ = 0.9914 (8)

The specific surface area of ZVI was 0.078 m^2^/g. The *k_SA_* values for ZVI dosages of 5, 10, 20, 30, and 40 g/L, obtained by converting *k_obs_* into the surface-area-normalized reaction rate, were 0.0833, 0.0832, 0.0678, 0.0614, and 0.0573 L/(h·m^2^), respectively. Contrasting with *k_obs_*, *k_SA_* decreased as the dosage increased, suggesting that sufficient active reaction sites were provided by the system when the dosage was over 20 g/L. Therefore, a ZVI dosage of 20 g/L was used in the subsequent experiments.

Figure 6 reveals that the relative concentration of CHCl_3_ increases initially and then subsequently decreases; the peak time occurred earlier as the ZVI dosage increases. When ZVI dosages were 30 and 40 g/L, peak CHCl_3_ concentrations occurred at 16 h. Conversely, when the ZVI dosage was 10 g/L, the peak CHCl_3_ concentration occurred at 48 h and when ZVI was 5 g/L, the peak CHCl_3_ concentration occurred after 72 h. However, experimental results also showed that the peak concentration of CHCl_3_ did not change significantly with changes in ZVI dosage with the range of 0.592–0.637. The ratio of CCl_4_ transformed to CHCl_3_ varied within the range of 0.729–0.763 under different ZVI dosages, indicating that a varied dosage only changed the CCl_4_ degradation rate, while the degradation pathway of CCl_4_ by ZVI was not altered.

### 3.4. Effect of Initial Solution pH on Reductive Dechlorination of CCl_4_

The pH of the reaction solution can promote corrosion or passivation of ZVI, and the redox half-reaction of chlorinated hydrocarbons (RCl + 2e^−^ + H^+^→RH + Cl^−^, *E*^0^ = 0.5–1.5 V when pH = 7) indicates that H^+^ is directly involved in the reductive dechlorination of chlorinated hydrocarbons. Therefore, pH is an important factor influencing the reductive dechlorination of CCl_4_ by ZVI. As described in Figure 7, pH had a considerable influence on CCl_4_ removal. After 8 h, CCl_4_ removal rates at pH values of 5, 6, 7, and 8 were 72.1%, 69.6%, 59.7%, and 50.0%, respectively, and after 48 h CCl_4_ removal rates exceeded 99.0% in all cases. Jiao et al. showed that the lower pH accelerated the dechlorination reaction of CCl_4_ with ZVI [30]. Shih et al. reported that decreasing pH from 9.3 to 3.2 resulted in an increase in pseudo first-order reaction constants of hexachlorobenzene by nZVI from 0.052 h^−1^ to 0.12 h^−1^ [31]. Figure 8 shows the pseudo first-order reaction kinetics of CCl_4_ degradation at different pH values. It was apparent that the degradation of CCl_4_ by ZVI at different pH values corresponded to pseudo first-order reaction kinetics. The *k_obs_* decreased with increasing pH, with the trend line fitted to Equation (9).
*k_obs_* = −0.0239 *pH* + 0.2837*, R*^2^ = 0.9684 (9)

The influence of initial pH on the reductive dechlorination of CCl_4_ can be attributed to the participation of H^+^ during the reaction. Moreover, lower solution pH is linked to accelerated ZVI corrosion [32] and prevention of the formation of passivation layer on ZVI surface [33]. The electrochemical corrosion process of ZVI in the solution can be expressed as follows [34]:

Under acidic conditions:2*Fe* + 4*H*^+^_(*aq*)_ + *O*_2(*aq*)_→2*Fe*^2+^ + 2*H*_2_*O*, *E* = 1.67 V(10)
4*Fe*^2+^ + 4*H*^+^_(*aq*)_ + *O*_2(*aq*)_→4*Fe*^3+^ + 2*H*_2_*O*, *E* = 0.46 V (11)

Under neutral and alkaline conditions:2*Fe* + 2*H*_2_*O*→2*Fe*^2+^ + *H*_2(*g*)_ + 2*OH*^−^, *E* = −0.39 V (12)
2*Fe*^2+^ + 2*H*_2_*O*→2*Fe*^3+^ + *H*_2(*g*)_ + 2*OH*^−^, *E* = −1.60 V (13)

Therefore, under weakly acidic conditions, sufficient H^+^ could be provided by the system for ZVI corrosion, leading to the steady generation of H_2_ for CCl_4_ degradation [35]. Under low pH conditions, ferrous hydroxide and other passivation films were also prevented from accumulating on ZVI surfaces, thereby creating more active reaction sites [36]. Moreover, low pH also promoted Fe^2+^ release and adsorption on the ZVI surfaces. The Fe^2+^ adsorbed on the mineral surfaces promoted the transfer of electrons from ZVI to CCl_4_, which in turn promoted the reductive dechlorination of CCl_4_ [37]. Under alkaline conditions, a series of complex iron hydroxide ions (such as [Fe(OH)]^+^, [Fe(OH)_3_]^−^, and [Fe(OH))_4_]^2−^) may have been formed on the ZVI surfaces, thereby inhibiting the reductive dechlorination reaction.

### 3.5. Significance of the Experimental Results on Permeable Reactive Barrier Design

One of the key factors considered in permeable reactive barrier (PRB) design is the residence time required for contaminants to reach the target concentrations, which generally determine the thickness of the PRB. Because contaminant degradation occurs within the PRB, in addition to considering the concentration of the target contaminants, it is also necessary to consider the degradation pathways of the contaminants and their intermediate products when determining the appropriate residence time. The required thickness of the PRB can be estimated by Equations (14) and (15) using the reaction rate constants of contaminant degradation, groundwater flow rates of the contaminated site, and the control concentrations of the target contaminants [38].
(14)$${b=v\;\times \;t}_{ref}\;\times SF$$
(15)$$t_{res}=\frac{-ln\frac{C_{t}}{C_{0}}}{k_{obs}}$$
where *b* is the PRB thickness, *v* is the flow rate of groundwater in the reaction medium, *t_res_* is the residence time of the contaminant, *SF* is the safety factor, *C*_0_ is the initial contaminant concentration, *C_t_* is the target contaminant concentration downstream of the PRB, and *k_obs_* is the reaction rate constant of contaminant degradation.

Supposing that the initial concentrations of CCl_4_ and CHCl_3_ in the groundwater in a specific contaminated site were 1000.0 μg/L and 0.0 μg/L, respectively, the values for the transformation parameters of CCl_4_ and CHCl_3_ in the ZVI-H_2_O system could be obtained from the results in Table 1. Using Equations (3)–(7), the concentration of CCl_4_ and CHCl_3_ for different residence times are described in Figure 9. After 27 h, CCl_4_ concentration decreased from 1000.0 μg/L to 57.0 μg/L, while CHCl_3_ concentration increased from 0.0 μg/L to 460.5 μg/L before decreasing gradually. According to China’s Standards for Drinking Water Quality (GB5749-2006), the maximum allowable concentrations of CCl_4_ and CHCl_3_ are 2.0 μg/L and 60.0 μg/L, respectively. In order to achieve the control targets, if CCl_4_ was considered on its own, the minimum residence time of contaminants in the PRB would be 59 h, and the concentrations of CCl_4_ and CHCl_3_ would be 1.9 μg/L and 377.2 μg/L, respectively. If CCl_4_ and CHCl_3_ were considered simultaneously, the minimum residence time would be 277 h, and the concentration of CHCl_3_ would be 59.8 μg/L, which was lower than the maximum allowable concentration. Therefore, the generation and degradation of CHCl_3_ would be the determining factor when determining minimum residence time.

## 4. Conclusions

Sponge ZVI particles can effectively degrade CCl_4_ in groundwater, which followed pseudo first-order reaction kinetics. The pseudo first-order reaction rate constant (*k_obs_*) of CCl_4_ was 0.1139 h^−1^ at a ZVI dosage of 20 g/L and an initial CCl_4_ concentration of 20 mg/L. During the degradation process, approximately 75% of CCl_4_ was transformed to CHCl_3_ by hydrogenolysis. The transformation of CHCl_3_ in the ZVI-H_2_O system was relatively slow, with the *k_obs_* of CHCl_3_ being 0.0109 h^−1^. Therefore, the minimum residence time was controlled by the production and subsequent transformation of CHCl_3_ for the ZVI-based PRB.

The main factors affecting CCl_4_ degradation by ZVI were the dosage of ZVI and the solution pH. CCl_4_ degradation was promoted when the ZVI dosage was increased and the *k_obs_* of CCl_4_ increased linearly with ZVI dosages from 0 to 40 g/L. The calculated surface-area-normalized reaction rate constants for the various dosages indicated the optimum ZVI dosage as 20 g/L. The weakly acidic conditions facilitated the degradation of CCl_4_, and the *k_obs_* of CCl_4_ decreased linearly for pH range of 5–8. Surface acidic washing had no significant effect on CCl_4_ removal by ZVI.

## Figures and Tables

**Figure 1 ijerph-18-12578-f001:**
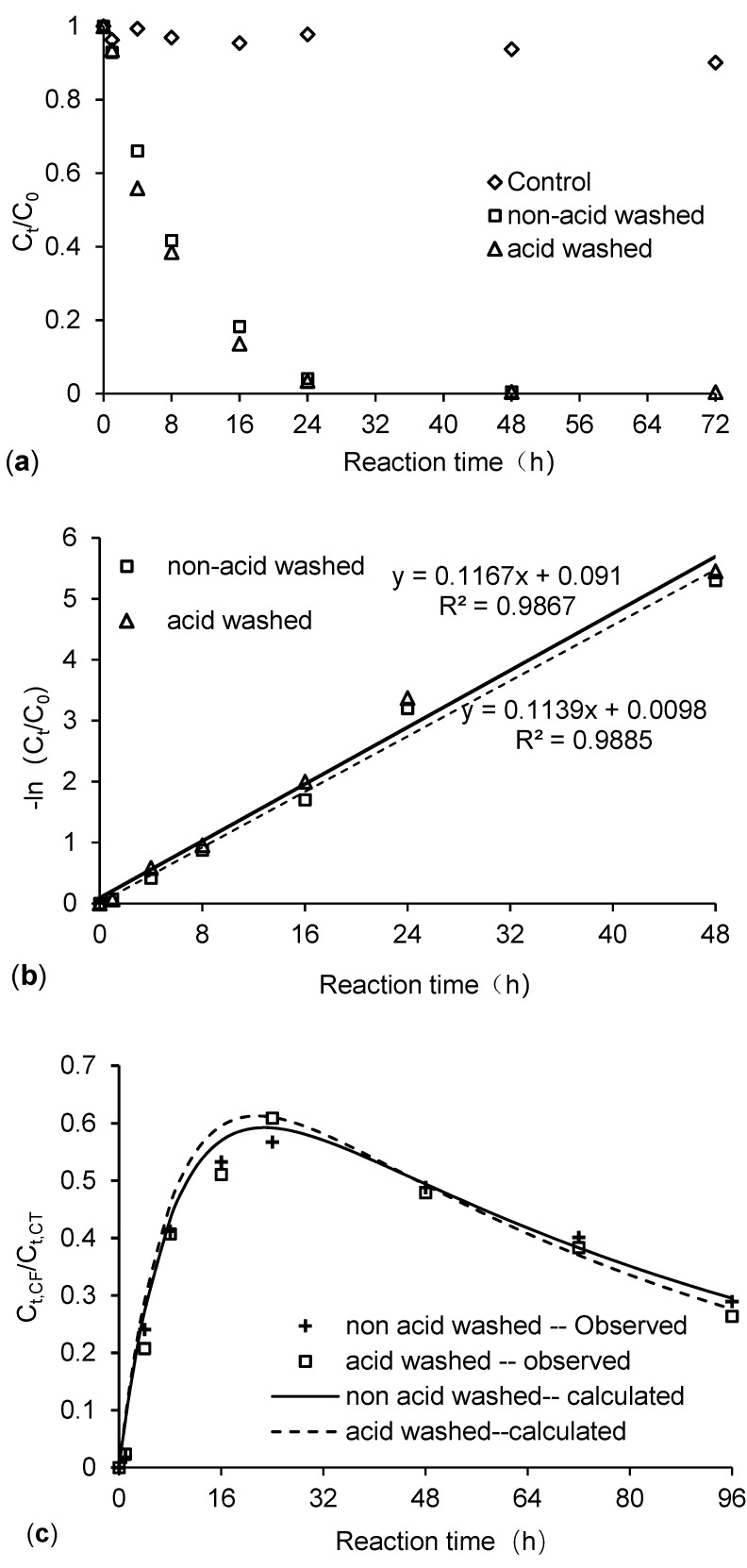
CCl_4_ degradation by acid-washed and non-acid washed ZVIs under the conditions of ZVI dosage of 20 g/L, initial CCl_4_ concentration of 20 mg/L, initial solution pH of 7 and stirring rate of 200 r/min. (**a**) Changes in CCl_4_ concentration with time, (**b**) degradation kinetics of CCl_4_. (**c**) Changes in concentration of chloroform with time. C_0_ was the initial CCl_4_ concentration, C_t_ was the CCl_4_ concentration at time t, and C_t,CF_ was the CHCl_3_ concentration at time t.

**Figure 2 ijerph-18-12578-f002:**
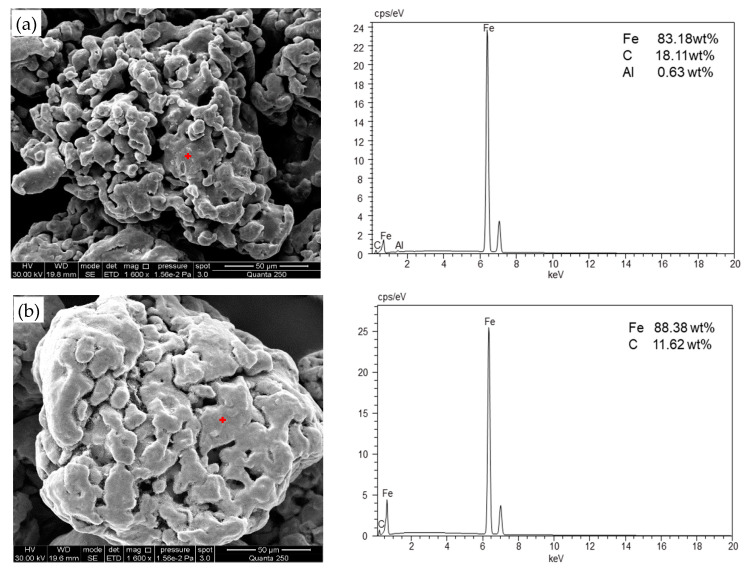
SEM images of ZVI: (**a**) acid-washed ZVI, (**b**) non-acid washed ZVI.

**Figure 3 ijerph-18-12578-f003:**
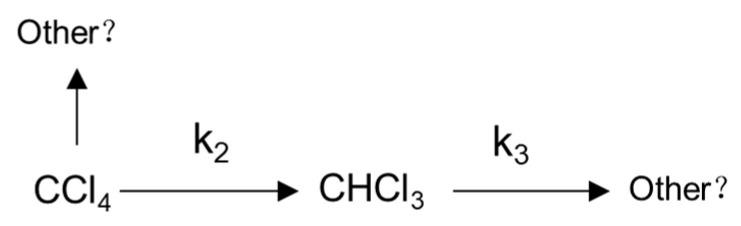
Reductive degradation mechanism of CCl_4_.

**Figure 4 ijerph-18-12578-f004:**
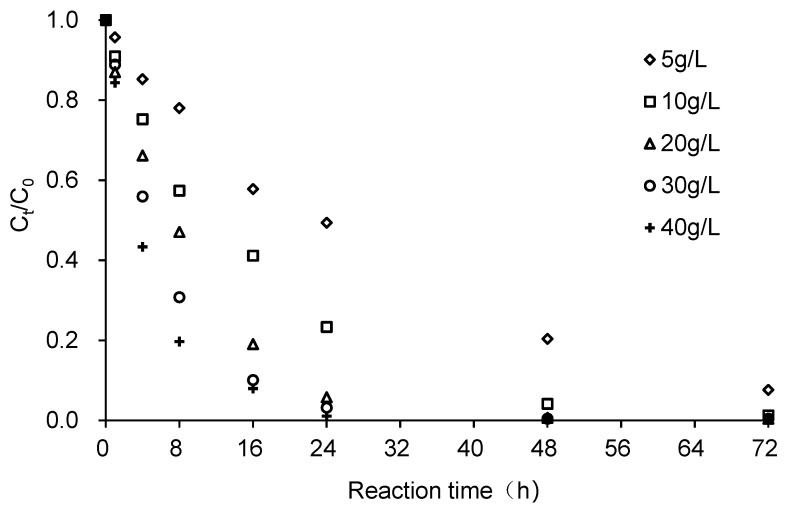
Effect of ZVI dosage on the degradation of CCl_4_. Reaction conditions were: ZVI dosage of 5, 10, 20, 30, and 40 g/L, CCl_4_ initial concentration of 20 mg/L, initial solution pH of 7 and stirring rate of 200 r/min. C_0_ was the initial CCl_4_ concentration and C_t_ was the CCl_4_ concentration at time t.

**Figure 5 ijerph-18-12578-f005:**
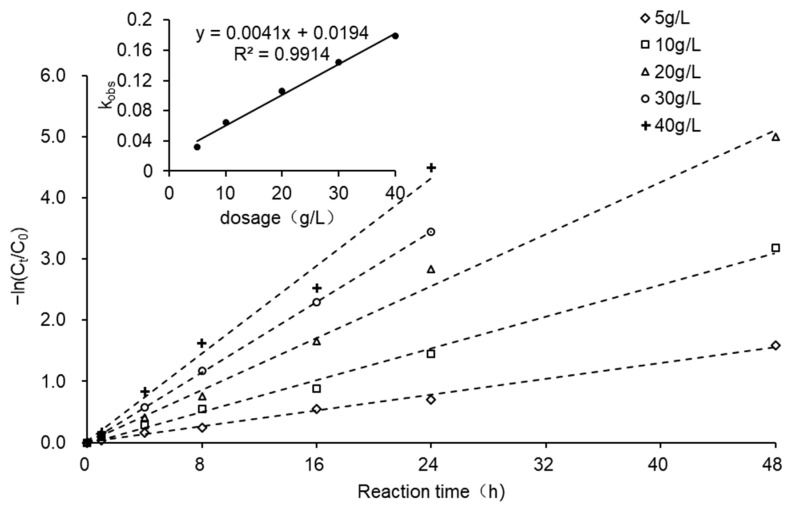
Reaction kinetics for CCl_4_ reductive dechlorination by ZVI with different dosages.

**Figure 6 ijerph-18-12578-f006:**
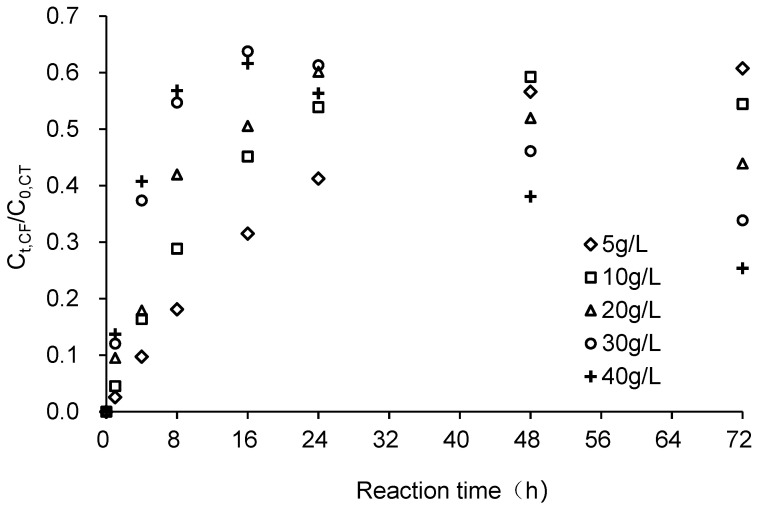
Changes in CF concentration at different dosages of ZVI. C_0,CT_ was the initial CCl_4_ concentration and C_t,CF_ was the CHCl_3_ concentration at time t.

**Figure 7 ijerph-18-12578-f007:**
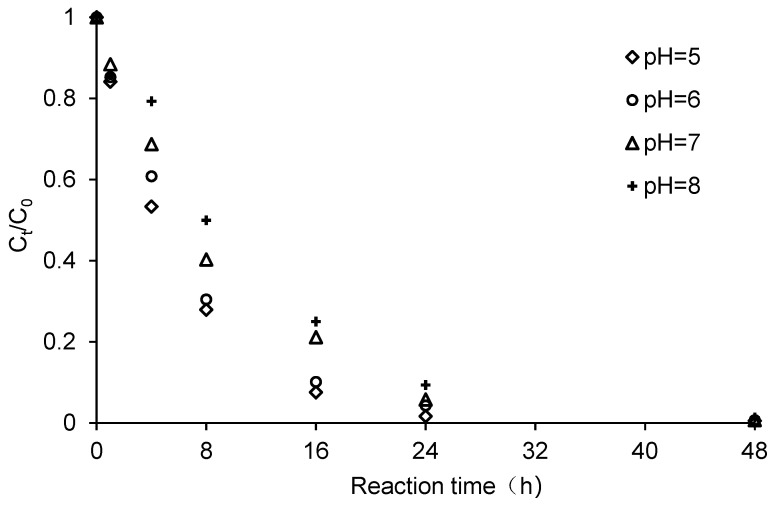
Effect of pH on CCl_4_ degradation by ZVI. Reaction conditions were: Initial solution pH of 5, 6, 7 and 8, ZVI dosage of 20 g/L, CCl_4_ initial concentration of 20 mg/L, stirring rate of 200 r/min. C_0_ was the initial CCl_4_ concentration and C_t_ was the CCl_4_ concentration at time t.

**Figure 8 ijerph-18-12578-f008:**
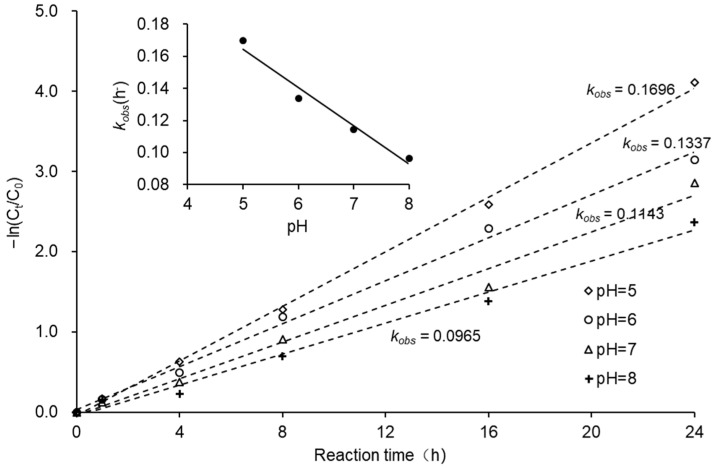
Reaction kinetics for CCl_4_ degradation at different initial solution pH.

**Figure 9 ijerph-18-12578-f009:**
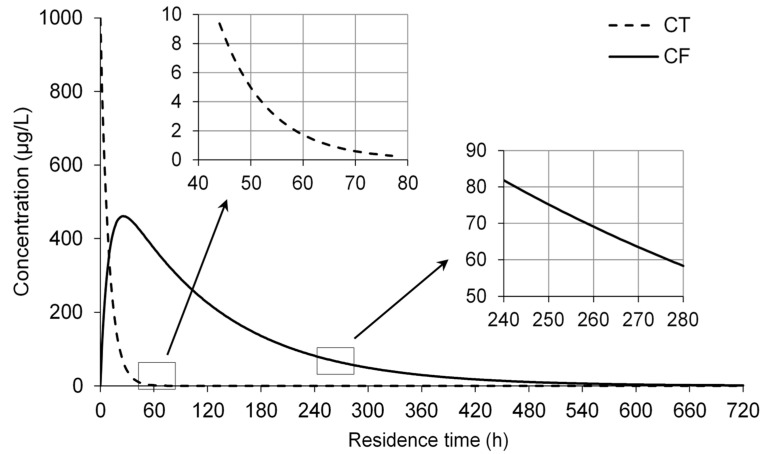
Changes in CCl_4_ concentration and its degradation products in the ZVI-water system. Assuming that initial CCl_4_ and CHCl_3_ concentrations were 1000 µg/L and 0.0 µg/L and k_obs,CT_ and k_obs,CF_ were 0.1139 h^−1^ and 0.0109 h^−1^, respectively.

**Table 1 ijerph-18-12578-t001:** The apparent reaction rate constants for CCl_4_ and chloroform degradation by acid and non-acid washed ZVI.

Reaction Medium	*k*_1_ (h^−1^)	*k*_2_ (h^−1^)	*k*_3_ (h^−1^)	*α*
Acid washed ZVI	0.1167	0.0931	0.0122	0.7980
Non-acid washed ZVI	0.1139	0.0865	0.0109	0.7569

## Data Availability

The data presented in this study are available on request from the corresponding author.

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
