# Peer review of "Degradation Characteristics of Carbon Tetrachloride by Granular Sponge Zero Valent Iron"

_ijerph, 2021, doi:10.3390/ijerph182312578_

Round 1

Reviewer 1 Report

This is a nice, carefully performed and clearly presented work. It merits publication.

A minor remark:

Section 2.1. The origin of CCL4 should be given. In addition, it is said "100 μL of 2×104 mg/L CCl4". Was a CCl4 solution used? What was the solvent?

Author Response

Comment 1: Section 2.1. The origin of CCl4 should be given. In addition, it is said "100 μL of 2×104 mg/L CCl4". Was a CCl4 solution used? What was the solvent?

Response 1: Two sentences were added in section 2.1. “CCl4 was analytical reagent grade and purchased from the Sinopharm Chemical Reagent Company (Shanghai, China).” “12.5 μL CCl4 was dissolved 1 mL methanol and then diluted to 10 mL with deionied water to prepare 2×104 mg/L CCl4”.

Reviewer 2 Report

This is an innovative and interesting study which reports the use of sponge Zero Valent Iron (ZVI) particles to degrade CCl4 in groundwater.  Reaction followed pseudo first-order reaction kinetics. The degradation process was effective and transformed approximately 75% of CCl4 to CHCl3. Further transformation of CHCl3 with ZVI was found to be relatively
slow.
Overall, the main factors controlling the degradation of CCl4 were the relative amounts of ZVI and the pH of the reaction solution.

This was a carefully planned and performed experimental study and the resulting paper is clear.  As far as I can see, it is suitable for publication as submitted.

Author Response

Comment: This is an innovative and interesting study which reports the use of sponge Zero Valent Iron (ZVI) particles to degrade CCl4 in groundwater. Reaction followed pseudo first-order reaction kinetics. The degradation process was effective and transformed approximately 75% of CCl4 to CHCl3. Further transformation of CHCl3 with ZVI was found to be relatively slow. Overall, the main factors controlling the degradation of CCl4 were the relative amounts of ZVI and the pH of the reaction solution.

This was a carefully planned and performed experimental study and the resulting paper is clear. As far as I can see, it is suitable for publication as submitted.

Response: Thank you for your affirmation.

Reviewer 3 Report

Zhu et al report a comprehensive study on the different parameters affecting granular sponge zero valent iron (ZVI) to degrade and remove carbon tetrachloride (CCl4). Thus, they rigorously evaluate the effect of a pre-wash with mineral acids, the dosage of ZVI and the effect of the initial pH, establishing which conditions are the most efficient. They also determine the kinetics of the process. In addition, they determine the time it would take for ZVI to be effective in a permeable reactive barrier, finding that it is very effective for the removal of CCl4 but not so effective for the removal of CHCl3, one of the reaction by-products. This work is of great interest to researchers working in the field of removing environmental pollutants such as CCl4 using metal particles.

Thus, it requires minor revisions in order to meet the journal's requirements.

Have the authors considered the possibility of combining ZVI with other metallic particles to remove both CCl4 and CHCl3, a reaction by-product, in a shorter time?

  • In the title of the manuscript, the word "valent" should begin with a capital letter.
  • Authors should check the writing: when describing results shown in tables or figures, the present tense is generally used as the verb tense. As an example, in line 201, it should be "Figure 5 shows" and not "Figure 5 showed". Please bear this in mind for all other cases.
  • Authors should carefully check the spacing between words and before almost all references. For example, in lines 30, 32, 35, 37... a space is missing between the word and the reference.
  • The quotient format (x/y) is used in the manuscript for units. However, it is preferable to use the negative exponent format (x y-1), because it is recommended by the IUPAC.
  • The paragraph in lines 52-58 could be better written, for clarity, as well as the sentence in lines 67-69
  • In line 70, (…) particles were washed, not was washed.
  • In line 71, (…) beaker by mixing and stirring.
  • Authors should check the correspondence between the citation number and the reference to which it refers. Thus, on line 130, citation 22 appears and then Phenrat et al, 2019, which is listed as reference no. 23. On the same line, the next sentence begins with Xin et al (2015) and ends by referring to reference 23, which is corresponds to Phenrat et al. In addition, they should follow the citation style of the journal, so they should check lines 130, 252 and 253.
  • The size of the figures must be the same in all cases.
  • Figures 1 and 3 could be combined into a single figure to better show the evolution of CCl4 and CHCl3.
  • In line 227, all the values of the constants should have the same number of decimal figures.
  • In Figures 7 and 10, the first sentence of the caption are in bold.
  • In Figure 9, the word “kinetics” of the caption begins with a capital letter.
  • In References’ section, author missed the bold letter of the year (ref. 24 and 25).

Author Response

Comment 1: Have the authors considered the possibility of combining ZVI with other metallic particles to remove both CCl4 and CHCl3, a reaction by-product, in a shorter time?

Response 1: Yes, we did the word that using Ag/Fe Bimetallic Particles to remove CCl4. The degradation rate of CCl4 in the accelerated reaction stage was 2.29–5.57-fold faster than that in the slow reaction stage, and 57.65–66.15-fold faster than that in the sponge ZVI. The paper was published in International Journal of Public Health and Environmental Research, 2021,18,2124.

Comment 2: In the title of the manuscript, the word "valent" should begin with a capital letter.

Response 2: The “valent” begins with a capital letter as suggested.

Comment 3: Authors should check the writing: when describing results shown in tables or figures, the present tense is generally used as the verb tense. As an example, in line 201, it should be "Figure 5 shows" and not "Figure 5 showed". Please bear this in mind for all other cases.

Response 3: We revised as suggested.

Comment 4: Authors should carefully check the spacing between words and before almost all references. For example, in lines 30, 32, 35, 37... a space is missing between the word and the reference.

Response 4: A space was added between words and references.

Comment 5: The quotient format (x/y) is used in the manuscript for units. However, it is preferable to use the negative exponent format (x y-1), because it is recommended by the IUPAC.

Response 5: The quotient format (x/y) is still used in the manuscript for units.

Comment 6: The paragraph in lines 52-58 could be better written, for clarity, as well as the sentence in lines 67-69

Response 6: The sentences in line 52-58 “In addition, it is reported that mZVIs was more effective for the removal of chlorinated aliphatic hydrocarbons compared to nZVIs when the ZVI material used singly for field applications according to 112 field case studies provided in the literatures[13]. Therefore, less expensive mZVI particles have been recently developed as a substitute for nZVIs to effectively degrade chlorinated hydrocarbons because of the high cost of nZVI and reduced lifetime due to high surface reactivity and fast corrosion rate with water[13-15]” was replace with “In addition, mZVIs removed chlorinated aliphatic hydrocarbons more effectively than nZVIs when the ZVI material used singly for field applications according to 112 field case studies provided in the literatures [13]. Recently, less expensive mZVI particles have been recently developed as a substitute for nZVIs to effectively degrade chlorinated hydrocarbons because of the high cost of nZVI and shortened lifetime due to high surface reactivity and fast corrosion rate with water [13-15].”

The sentence in lines 67-69 “The material had a mean grain size less than 150 µm and a specific surface area of 0.078 m2/g [16] and was also analyzed for Fe (96.52%), O (2.4%), Si (0.5%), Mn (0.29%), Ca (0.13%), Cr (0.048%), Mg (0.043%), P (0.038%), and S (0.031%) using S8-TIGER X-ray Fluorescence (Bruker Corporation, Germany)” was replace with “The material had a mean grain size less than 150 µm and a specific surface area of 0.078 m2/g [16] and the composition of the ZVI incluing Fe (96.52%), O (2.4%), Si (0.5%), Mn (0.29%), Ca (0.13%), Cr (0.048%), Mg (0.043%), P (0.038%), and S (0.031%) was also analyzed using S8-TIGER X-ray Fluorescence (Bruker Corporation, Germany)”

Comment 7: In line 70, (…) particles were washed, not was washed.

Response 7: Revised as suggested.

Comment 8: In line 71, (…) beaker by mixing and stirring.

Response 8: Revised as suggested.

Comment 9: Authors should check the correspondence between the citation number and the reference to which it refers. Thus, on line 130, citation 22 appears and then Phenrat et al, 2019, which is listed as reference no. 23. On the same line, the next sentence begins with Xin et al (2015) and ends by referring to reference 23, which is corresponds to Phenrat et al. In addition, they should follow the citation style of the journal, so they should check lines 130, 252 and 253.

Response 9: We checked the citation number and the reference through the manuscript and did some corrections.

Comment 10: The size of the figures must be the same in all cases.

Response 10: We adjusted the size of some figures as suggested.

Comment 11: Figures 1 and 3 could be combined into a single figure to better show the evolution of CCl4 and CHCl3.

Response 11: Figure 3 was combined into Figure 1 as suggested.

Comment 12: In line 227, all the values of the constants should have the same number of decimal figures.

Response 12: Revised as suggested.

Comment 13: In Figures 7 and 10, the first sentence of the caption are in bold.

Response 13: Revised as suggested.

Comment 14: In Figure 9, the word “kinetics” of the caption begins with a capital letter.

Response 14: Revised as suggested.

Comment 15: In References’ section, author missed the bold letter of the year (ref. 24 and 25).

Response 15: Revised as suggested.
